# Influencing Factors on Intersegmental Identification Adequacy in Segmentectomy with Intraoperative Indocyanine Green (ICG) Intravenous Administration

**DOI:** 10.3390/cancers15245876

**Published:** 2023-12-17

**Authors:** Harushi Ueno, Tomohiro Setogawa, Ayaka Makita, Yuko Ohara, Yoshito Imamura, Shoji Okado, Hiroki Watanabe, Yuta Kawasumi, Yuka Kadomatsu, Taketo Kato, Shota Nakamura, Tetsuya Mizuno, Toyofumi Fengshi Chen-Yoshikawa

**Affiliations:** Department of Thoracic Surgery, Nagoya University Graduate School of Medicine, Nagoya 466-8560, Japan; setogawa.t1@med.nagoya-u.ac.jp (T.S.); yuuko_oohara@med.miyazaki-u.ac.jp (Y.O.); y-imamura@med.nagoya-u.ac.jp (Y.I.); s-okado@med.nagoya-u.ac.jp (S.O.); halcyon_days_19@yahoo.co.jp (H.W.); ykawasumi@med.nagoya-u.ac.jp (Y.K.); ykadomatsu@med.nagoya-u.ac.jp (Y.K.); tkato@med.nagoya-u.ac.jp (T.K.); shota197065@med.nagoya-u.ac.jp (S.N.); te.mizuno@med.nagoya-u.ac.jp (T.M.); tyoshikawa@med.nagoya-u.ac.jp (T.F.C.-Y.)

**Keywords:** Indocyanine Green (ICG), robot-assisted thoracoscopic surgery (RATS), video-assisted thoracoscopic surgery (VATS), segmentectomy

## Abstract

**Simple Summary:**

Segmentectomy, a surgical technique aimed at preserving as much lung tissue as possible, has become increasingly common for many small lung cancers. Since the boundaries of lung regions cannot be directly identified, various methods are employed to visualize them. The method involves using Indocyanine Green (ICG). During surgery, the vessels supplying the area to be removed are dissected, followed by the intravascular administration of ICG. When viewed with a near-infrared camera, the remaining lung fluoresces, enabling clear visualization of region boundaries. This technique is both convenient and safe. Despite following the correct procedure, there may still be instances where the region boundaries are not clearly delineated. In such instances, it is crucial to anticipate the need for alternative methods to define regional boundaries in advance. In this study, we conducted retrospective research to ascertain the causes of the unclear boundaries.

**Abstract:**

Accurate identification of the intersegmental plane is essential in segmentectomy, and Indocyanine Green (ICG) assists in visualizing lung segments. Various factors, including patient-related, intraoperative, and technical issues, can influence boundary delineation. This study aims to assess the rate of unsuccessful intersegmental identification and identify the contributing factors. We analyzed cases of lung segmentectomy from April 2020 to March 2023, where intraoperative ICG was intravenously administered during robot-assisted or video-assisted thoracoscopic surgery. Cases where fluorescence extended beyond expected boundaries within 30 s were classified as the “unclear boundary group”. This group was then compared to the “clear boundary group”. The study encompassed 111 cases, 104 (94%) of which were classified under the “clear boundary group” and 7 (6%) under the “unclear boundary group”. The “unclear boundary group” had a significantly lower DLCO (15.7 vs. 11.8, *p* = 0.03) and DLCO/VA (4.3 vs. 3.0, *p* = 0.01) compared to the “clear boundary group”. All cases in the “unclear boundary group” underwent lower lobe segmentectomy. ICG administration effectively outlines pulmonary segments. Challenges in segment demarcation may occur in cases with low DLCO and DLCO/VA values, particularly during lower lobe segmentectomy.

## 1. Introduction

Recent advancements in radiology and surgery have facilitated the detection and minimally invasive resection of small lung cancers globally [1]. The results of the JCOG0802 and JCOG1211 trials, combined with the introduction of robot-assisted thoracic surgery (RATS) into insurance coverage from April 2020, are expected to lead to an increase in lung segmentectomy cases for small lung cancers in Japan [2,3]. Segmentectomy has the advantage of preserving the lung function compared to lobectomy, but since the boundaries between segments are not visually perceptible, they need to be visualized. If accurate intersegmental dissection is not performed, there is a risk of the tumor not being included in the lung to be excised, insufficient tumor margins, inadvertent cutting into the tumor, or ischemia of the residual lung due to the incorrect site of resection. Therefore, identifying the intersegmental plane is a crucial procedure in segmentectomy [4,5,6,7,8,9,10,11,12]. Traditionally, the process of identifying intersegmental boundaries involved blocking the bronchi within the resection area and determining the inflation-deflation line. However, the efficacy of this method in achieving clear intersegmental delineation, particularly in severe emphysema cases, is limited due to the expansion of the inflation–deflation line through Kohn’s alveolar pores [7].

Indocyanine Green (ICG) is a compound that binds to plasma proteins and fluoresces when exposed to excitation light. This property can identify intersegmental boundaries. Specifically, after resecting the dominant pulmonary artery and vein in the resection area, ICG is administered intravascularly. Observing the lung with near-infrared thoracoscopy enables the blood-containing ICG to flow into and fluoresce the lung that requires preservation. This method enables the accurate identification of intersegmental boundaries through blood flow control. Even with this method, there are instances where intersegmental boundaries are not clearly defined.

The failure of intersegmental delineation is due to operator-related factors, such as misidentifying blood vessels for resection and improper equipment usage. Patient-related factors can contribute to unclear boundaries, even with proper blood vessel dissection and intravascular ICG administration. During surgery, we encountered instances where ICG fluorescence extended beyond the expected region into the resection area, hindering intersegmental delineation, despite accurate blood vessel dissection and intravascular ICG administration. We conducted a retrospective study at our institution to identify the causes of these occurrences. The study involved lung segmentectomy with intraoperative ICG intravenous administration during video-assisted thoracic surgery (VATS) or RATS.

## 2. Materials and Methods

We conducted a retrospective study on cases where intraoperative ICG was administered intravenously for intersegmental identification during VATS or RATS segmentectomy for malignant lung tumors at our institution, from April 2020 to March 2023. Cases involving prior same-side lung surgery were excluded. The data were extracted from our department’s database, and any missing items were complemented using information from hospital medical records. Consequently, there were no instances of missing values in any of the cases.

The study encompassed a total of 111 cases (Table 1). Among them, 104 cases (94%) belonged to the clear boundary group, while 7 cases (6%) were part of the unclear boundary group. The median age was 70 years, with males making up 55% of the individuals. The median of the Charlson Comorbidity Index (CCI) was 1. Smokers constituted 60% of the cases. The median vital capacity (VC) was noted to be 3280 mL, with a corresponding median %VC of 106.5%. The median forced expiratory volume in one second (FEV1) was 2370 mL, with a corresponding median %FEV1 of 100.4%. The median lung diffusing capacity for carbon monoxide (DLCO) was 15.64 mL/min/mmHg, and the median ratio of DLCO to vital capacity (DLCO/VA) was 4.22 mL/min/mmHg/L. In the cases, 32% demonstrated an obstructive pattern of ventilatory impairment. The median maximum tumor size measured 17 mm. RATS was conducted in 48% of cases, and VATS was performed in 52% of cases. Among these cases, 77% were diagnosed as lung cancer, and 23% were identified as metastatic lung tumors. The surgical procedure, simple segmentectomy, constituted 79%. The median dose of ICG administered was 7.5 mg, with a range of 5 mg to 20 mg. The median blood loss during the operation was 5 mL, and the median surgical duration was 155 min. No cases required conversion to open thoracotomy. Nine cases reported Clavien–Dindo grade 3 or higher postoperative complications, with a median post-surgery drainage duration of 2 days and a median hospital stay of 5 days post surgery.

In all instances, a 3-dimensional computed tomography (3D-CT) was preoperatively created using SYNAPSE VINCENT^®^ (Fujifilm Medical Co., Tokyo, Japan) to confirm the locations of the pulmonary artery, vein, and bronchus to be resected and to ensure adequate tumor margins. During the surgery, all pulmonary arteries, veins, and bronchi associated with the resected area were dissected. An anesthesiologist subsequently dissolved 25 mg of ICG in 10 mL of normal saline and administered it intravenously as directed by the supervising surgeon. In RATS, the Da Vinci Xi system^®^ (Intuitive Surgical Inc., Sunnyvale, CA, USA) was employed for all cases, with surgeries conducted using CO_2_ insufflation at −8 cmH2O. ICG fluorescence was observed in Firefly mode, with CO_2_ insufflation maintained throughout the process. In VATS, a 10 mm scope diameter was utilized, using either the IMAGE 1 S™ D-LIGHT P SCB, RUBINA™ (Karl Storz, Berlin, Germany), or VISERA ELITE II OTV-S300 (Olympus, Tokyo, Japan) to identify ICG fluorescence. Cases where the ICG fluorescence surpassed the predicted region within 30 s of reaching the lung field, and where the surgeon could not identify intersegmental regions solely through ICG intravenous administration, were categorized as the “unclear boundary group” (Figure 1). We analyzed preoperative, intraoperative, and postoperative factors between the two groups to identify the characteristics of cases with unclear boundaries. Individuals, excluding the operating surgeon, reviewed the surgical videos to validate the unclear boundary assessments. We cross-referenced preoperative 3D-CT scans with surgical videos to ensure that all vessels intended for dissection were completely separated.

### Statistical Methods

Descriptive statistics are represented as the median and range for continuous variables and as frequency and percentage for categorical variables. A ROC curve was created to assess the accuracy in predicting the determination of clear and unclear boundaries. The area under the curve (AUC) and its 95% confidence interval were calculated, and the cutoff points were determined using the Youden Index. All statistical analyses were conducted using EZR version 1.63 (Saitama Medical Center, Jichi Medical University, Saitama, Japan). The analysis was performed using the Mann–Whitney U test and Fisher exact test. All *p*-values were two-tailed or two-sided, with significance set at *p* < 0.05.

## 3. Results

Figure 2 shows a list of the resected lung segments. The most commonly resected section on the right side was S6, followed by S7-10. The most frequent resection on the left side was S1-3, followed by S6 and S4 + 5. One case involved segmentectomy of both lower lobes.

Table 2 shows the comparison between the two groups. In DLCO (11.8 [8.1–18.3] vs. 15.7 [6.4–26.9], *p* = 0.03, and DLCO/VA 3.0 [1.7–4.3] vs. 4.3 [1.4–17.4], *p* = 0.01), the unclear boundary group had significantly lower values. No significant difference was noted between the two groups regarding other factors. The unclear boundary group consisted of four cases of right S6 segmentectomy (including one case with additional upper lobectomy), two cases of left S6 segmentectomy, and one case of right S7-10 segmentectomy (with additional left S8 + 9 segmentectomy). All cases involved lower lobe segmentectomy. Postoperative complications of Clavien–Dindo grade 3 or higher occurred in one case (14%) [empyema] within the unclear boundary group and in eight cases (8%) [four persistent air leak, empyema, postoperative bleeding, stroke, and residual lung torsion] within the clear boundary group. None of these complications were directly linked to ICG.

The ROC curve analysis identified a cutoff value of 15.01 for DLCO (AUC 0.735, 95% CI 0.547–0.923) and 4.25 for DLCO/VA (AUC 0.808, 95% CI 0.657–0.959) (Figure 3).

In the clear boundary group, two cases were identified (Case 1: RATS right upper lobectomy + right S6 segmentectomy; Case 2: RATS left S4 + 5 segmentectomy) in which the fluorescence in the resected area revealed a pulmonary artery that still needed resection during the surgery. In Case 1, a 1mm A2b artery, not identified in the preoperative 3D-CT, was detected due to the fluorescence of the dissected area during intersegmental delineation. The A2b artery was subsequently recognized and dissected before intersegmental division (Figure 4). In Case 2, the surgeon mistakenly excised only the A4a artery during the dissection of the mediastinal type A4a + b artery. It was found that a part of the resection area had fluoresced, indicating the presence of a residual A4b artery. This was excised before the division of the intersegmental parenchyma (Figure 5).

## 4. Discussion

The technique of demarcating intersegmental boundaries using intraoperative intravenous ICG administration was successful in most cases, with a 94% success rate. This is in line with previous studies that reported success rates ranging from 85% to 100% [13,14,15,16,17,18,19,20,21,22,23,24,25,26,27,28,29]. Our study revealed that 6% of cases had unclear intersegmental delineation, significantly associated with lower DLCO and DLCO/VA values. Respiratory function test results for VC, %VC, FEV, and %FEV showed no significant differences between the well-delineated and poorly delineated groups. Reports indicate that severe emphysema or anthracosis cases may have unclear intersegmental boundaries when using intravenous ICG administration [7,14,19,27]. Yet, no studies have identified specific factors from pulmonary function test results that could contribute to poor delineation. Conditions such as emphysema, pulmonary fibrosis, and pulmonary hypertension, associated with reduced DLCO and DLCO/VA, are challenging to accurately assess based solely on preoperative imaging findings or pulmonary function tests (VC, %VC, FEV, %FEV). Underlying pulmonary diseases may potentially modify the characteristics of blood flow on the lung’s surface, leading to blurred segment boundaries in cases exhibiting low DLCO and DLCO/VA values.

In this study, the unclear delineation in all cases was attributed to the expansion of the ICG fluorescence area within the resection area. This phenomenon was exclusively observed in lower lobe segment resection cases, predominantly in S6 segmentectomy. The lower lobes have a higher blood flow than the upper lobes, potentially leading to an increased influx of ICG via pulmonary capillaries, contributing to the expansion of ICG into the resection area. We empirically determined that marking incision lines on the lung parenchyma with an electrical scalpel, using the intersegmental boundaries identified with ICG, necessitates roughly 30 s after the ICG has reached the lung field and defined the boundaries. Therefore, we established a time frame of approximately 30 s as the threshold for cases where intersegmental boundaries can be clearly identified and classified as having clear boundaries. Mun et al. reported that fluorescence becomes observable roughly 30 s post intravenous administration, lasting for a median duration of 180 s [25]. Although there may have been variations in the time taken for fluorescence observation or its duration between the clearly and unclearly delineated groups, our retrospective study did not measure these time-related factors, thus we were unable to assess this aspect.

In traditional equipment, infrared fluorescent images in the chest cavity were observed using near-infrared thoracoscopy excitation light. This made it difficult to distinguish structures within the chest cavity against a monochrome background, as areas outside the fluorescent region appeared dark. Recently, devices utilizing the overlay method, which combines fluorescent and HI Vision images, have become increasingly popular. This method provides exceptional visibility, distinctly displaying the fine surgical field background and blood flow regions. The Firefly mode, a fluorescence feature integrated into the Da Vinci system, is not an overlay technique. Instead, it vividly and brightly illuminates non-fluorescent areas, simplifying the visualization of the overall shape of the lungs. The surgeon successfully identified unintended residual blood vessels in two instances, thereby enhancing the safety of the surgery. Conversely, no significant difference was observed in the interregional delineation success rate when compared to thoracoscopic surgery. In our facility, the near-infrared systems utilized for VATS were not standardized, hindering a comparison under identical conditions. It is believed that the success rate of interregional delineation between VATS and RATS is not affected by differences in their equipment. It was concluded that a CO_2_ insufflation of approximately −8 cmH2O had no significant impact on interregional delineation. Our results showed no significant difference in the success rate of intersegmental delineation based on the administered ICG dosage. Additionally, we observed no complications directly attributed to ICG administration. Previous reports from various institutions suggest that an intravenous dosage of ICG, approximately between 5 mg and 20 mg, is suitable [13,14,15,16,17,19,20,21,22,23,24,25,26,27,28,29]. Misaki et al. illustrated that the fluorescence intensity could be affected by the ICG administration method [30]. Improving the administration method could potentially expand the “clear boundary group”. However, no matter how refined the administration method may be, instances are likely to occur where intravenous administration of ICG fails to delineate intersegmental boundaries. Anticipating cases with potentially unclear boundaries and incorporating alternative approaches, such as intraoperative cone-beam computed tomography or virtual assisted lung mapping [1], to elucidate tumor location and determine resection areas are crucial procedures. Our findings are valuable in predicting unclear boundary cases.

This study has multiple limitations. This is primarily a retrospective study. Second, the limited sample size only allows for a basic comparison between the two groups. Third, despite consistent assessments from surgeons other than the operating one, the categorization into the “clear boundary group” and “unclear boundary group” is subjective. The intravenous administration of ICG for interregional delineation is deemed safe and effective. However, it requires further investigation and more case studies in the future.

## 5. Conclusions

In conclusion, our study has uncovered new findings indicating the potential for unclear intersegmental delineation in cases with low DLCO or DLCO/VA and in cases involving lower lobe segmentectomy. The intravenous administration of ICG for segmentectomy is, in most cases, a straightforward and effective method. Complications arising from ICG administration were not observed. Additionally, there were cases where the use of ICG enabled surgeons to safely perform surgeries. In cases with low DLCO or DLCO/VA, as well as cases involving lower lobe segmentectomy, it is crucial to anticipate that ICG administration may not provide a clear delineation of intersegmental areas. Therefore, preparing alternative methods is essential. Due to the high success rate of intersegmental delineation with ICG, there were few cases of inadequate delineation, which is a limitation of this study. To comprehensively understand the causes and develop effective strategies for such cases, further research and case accumulation are imperative.

## Figures and Tables

**Figure 1 cancers-15-05876-f001:**
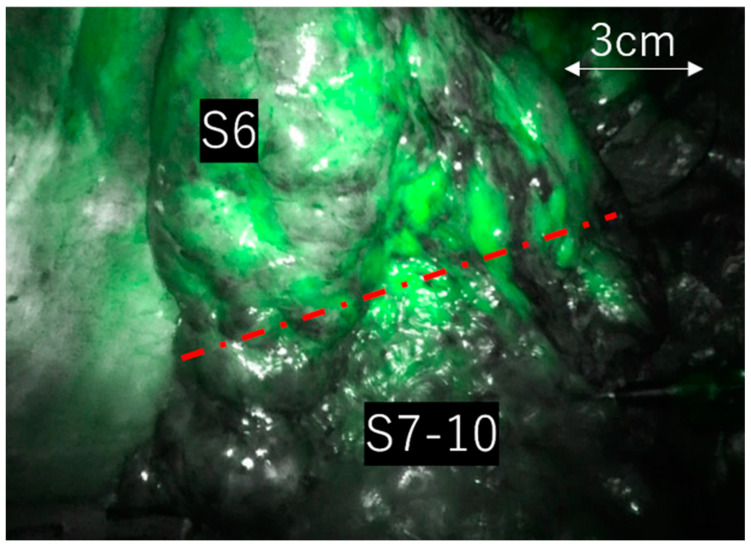
Case of right S6 segmentectomy. Even after all vessels were disconnected, the ICG fluorescence expanded beyond the boundary within 30 s, rendering segment differentiation impossible.

**Figure 2 cancers-15-05876-f002:**
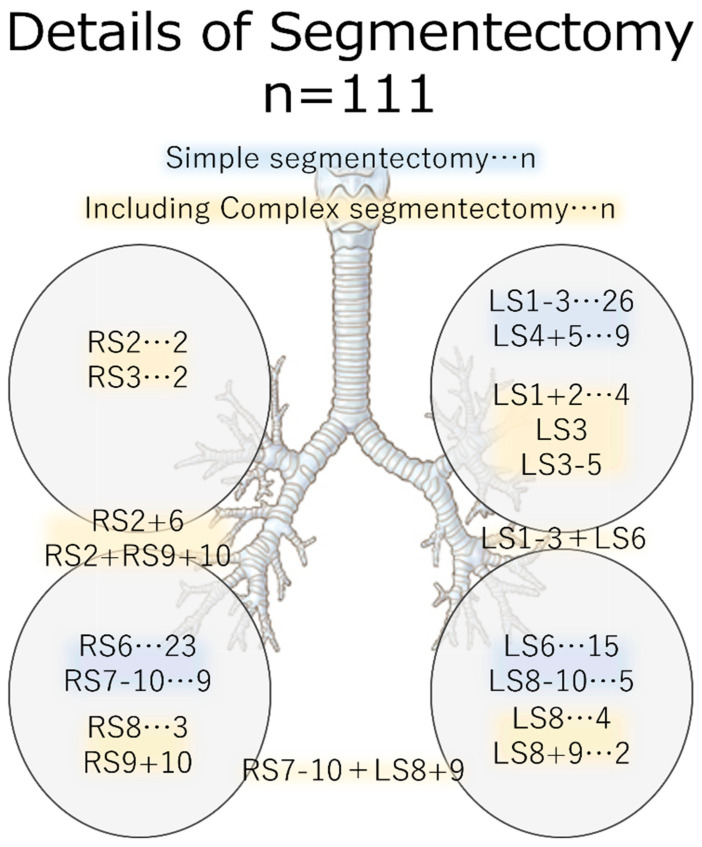
Detailed view of the resected area. Simple segmentectomy is highlighted in blue, while complex segmentectomy procedures are marked in yellow.

**Figure 3 cancers-15-05876-f003:**
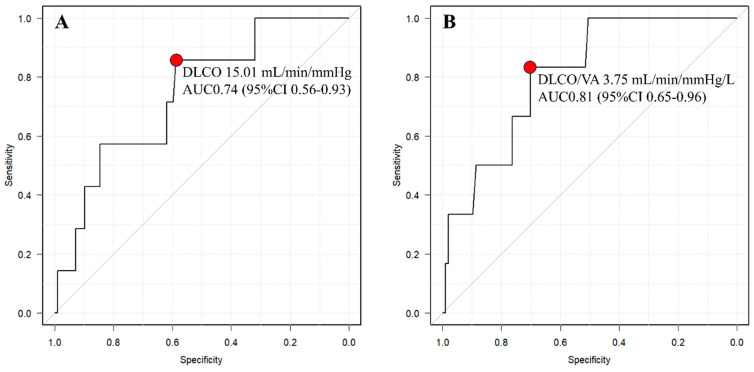
Receiver operating characteristic (ROC) curve between “clear boundary group” and “unclear boundary group” of DLCO (**A**) and DLCO/VA (**B**). The red points indicate the cutoff values. AUC, area under curve.

**Figure 4 cancers-15-05876-f004:**
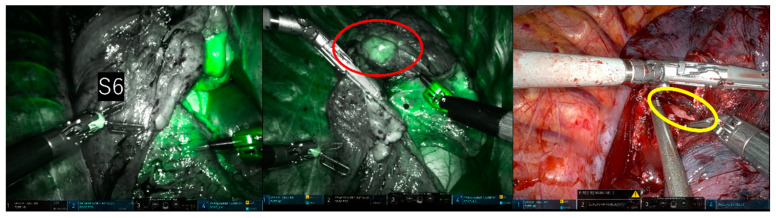
Upper right lobectomy and S6 segmentectomy. During intersegmental delineation, fluorescence (indicated by the red circle) was detected in the resection area due to a 1mm A2b artery (indicated by the yellow circle) not previously identified on preoperative 3D-CT. The non-fluorescent lung tissue, with its clear and bright outline, was easily identifiable as an upper lobe section to be excised.

**Figure 5 cancers-15-05876-f005:**
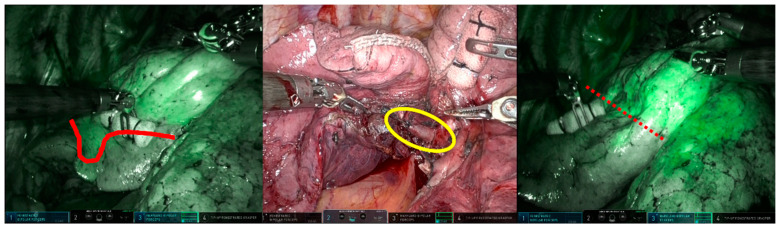
Segmentectomy of left S4 + 5. The surgeon mistakenly thought that they were resecting the mediastinal A4a + b artery. However, the designated boundary (marked by the red line) was too peripheral to serve as the boundary between S3 and S4 + 5. The A4b artery, indicated by the yellow circle, was intact. Cutting it confirmed the precise boundary (marked by the red dot line).

**Table 1 cancers-15-05876-t001:** Characteristics of patients included in this study.

Characteristic	Value
Clear boundary group/Unclear boundary group	104 (94)/7 (6)
Median Age (Range)	70 [22–91]
Sex, male (%)	61 (55)
Body Mass Index (Median [Range])	22.9 [15.0–30.8]
CCI (Median [Range])	1 [0–9]
Smoking Habit, N (%)	66 (60)
FVC, mL (Median [range])	3280 [1670–5120]
%FVC, % (Median [range])	106.5 [71.9–138.1]
FEV1, mL (Median [range])	2370 [1040–4200]
%FEV1,0% (Median [Range])	100.4 [44.4–137.5]
DLCO, mL/min/mmHg (Median [Range])	15.6 [6.4–26.9]
DLCO/VA, mL/min/mmHg/L (Median [Range])	4.2 [1.4–17.4]
Pattern of Ventilatory Impairment, *n* (%) Within normal limits Restrictive Obstructive Mixed	73 (66) 1 (1) 36 (32) 1 (1)
Clinical Maximum Tumor Size, mm (Median [Range])	17 [4–38]
Surgical Approach: RATS (%)	53 (48)
Lung Cancer (%)	85 (77)
Segmentectomy Type: Simple (%)	88 (79)
ICG Dose, mg (Median [Range])	7.5 [5–20]
Median Blood Loss, mL (Range)	5 [0–200]
Operation Duration, Minutes (Median [Range])	155 [77–360]
Postoperative Complications Grade > 3, *n* (%)	9 (8)
Postoperative Drainage Duration, Day (Median [Range])	2 [1–18]
Median Postoperative Hospital Stay (Days) [Range]	5 [3–28]

**Table 2 cancers-15-05876-t002:** Comparison of clear and unclear boundary groups.

Characteristic	Clear Boundary Group *n* = 104	Unclear Boundary Group (*n* = 7)	*p*-Value
Median Age (Range)	69.5 [22–85]	75 [47–91]	0.07
Percentage of Males (%)	56 (54)	5 (71)	0.46
Body Mass Index (Median [Range])	23.0 [15–31]	22 [17–24]	0.19
CCI (Median [Range])	1 [0–9]	1 [0–8]	0.97
Brinkman Index (Median [Range])	270 [0–2925]	360 [0–900]	0.84
VC, mL (Median [Range])	3150 [1670–5120]	3450 [2430–4420]	0.77
%VC, % (Median [Range])	106.0 [72.5–135.4]	109.9 [71.9–138.1]	0.21
FEV1, mL (Median [range])	2370 [1040–4200]	2450 [1610–3040]	0.64
%FEV1,0% (Median [Range])	100.2 [44.4–137.5]	110.7 [64.9–131.5]	0.26
DLCO, mL/min/mmHg (Median [Range])	15.7 [6.4–26.9]	11.8 [8.1–18.3]	0.03
DLCO/VA, mL/min/mmHg/L (Median [Range])	4.3 [1.4–17.4]	3.0 [1.7–4.3]	0.01
Pattern of Ventilatory Impairment (%) Within normal limits Restrictive Obstructive Mixed	68 (65) 0 35 (34) 1 (1)	5 (72) 1 (14) 1 (14) 0	0.08
Clinical Maximum Tumor Size, mm (Median [Range])	17 [4–38]	23 [14–37]	0.06
Surgical Approach: RATS (%)	50 (48)	3 (43)	1.00
Lung Cancer (%)	80 (77)	5 (71)	0.67
Segmentectomy Type: Simple (%)	82 (79)	6 (86)	1.00
ICG Dose, mg (Median [Range])	7.5 [5–20]	7.5 [5–10]	0.89
Median Blood Loss, mL (Range)	5 [0–105]	15 [0–200]	0.13
Operation Duration, Minutes (Median [Range])	155 [77–360]	199 [135–273]	0.08
Postoperative Complications Grade > 3, *n* (%)	8 (8)	1 (14)	0.46
Postoperative Drainage Duration, Day (Median [Range])	2 [1–18]	3.5 [2–10]	0.12
Median Postoperative Hospital Stay (Days) [Range]	5 [3–28]	6 [5–26]	0.16

## Data Availability

Raw data were generated at Department of Thoracic Surgery, Nagoya University Graduate school of Medicine, Nagoya, Japan. Derived data supporting the findings of this study are available from the corresponding author H.U. on request.

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
