# Peer review of "Influencing Factors on Intersegmental Identification Adequacy in Segmentectomy with Intraoperative Indocyanine Green (ICG) Intravenous Administration"

_cancers, 2023, doi:10.3390/cancers15245876_

Round 1

Reviewer 1 Report

Comments and Suggestions for Authors

This study aims to assess the rate of unsuccessful intersegmental identification and identify the contributing factors. They analyzed cases of lung segmentectomy from April 2020 to March 2023, where intraoperative Indocyanine Green was intravenously administered during robot-assisted or video-assisted thoracoscopic surgery.

In this manuscript, purpose, method, and results are well described and the obtained facts are well discussed. I think that this manuscript is a good candidate as a paper publishable in Cancers. I may suggest one revision. This manuscript provides important proposal, but description in Conclusion is so insufficient. More description on research impact and future perspective had better be added in Conclusion section.

Reviewer 2 Report

Comments and Suggestions for Authors

Referee Report

 This study conducted a retrospective analysis aiming to identify the reasons behind unclear boundaries when utilizing ICG in lung segmentectomy. Several concerns have been identified within this study:

  1. Authors' Affiliation: It is advisable to eliminate repeated affiliations listed within the document.
  2. Introduction: The authors should emphasize the significance of this study, emphasizing why the inability to accurately identify tumor boundaries is critical. Additionally, they might discuss alternative procedures involving different materials like nanomaterials to delineate tumor boundaries.
  3. Materials and Methods, Line 97: Clarification is needed regarding the exclusion of patients above 20 years of age from the patient group.
  4. Materials and Methods: While the authors presented patient data in the Results section along with Table 1, it would be more appropriate to include this information and Table 1 within the Materials and Methods section.
  5. Materials and Methods, Line 122: Please specify the complete form of EZR software used and its version.
  6. Results: The analysis is based on only 7 failed cases, which appears insufficient for a comprehensive analysis.
  7. Figure 1: Including a ruler bar in the photo would provide a scale reference for the tumor's size.
  8. Table Captions: Missing captions need to be added for both Table 1 and Table 2.
  9. Discussion: Given the results obtained, the authors should propose potential solutions to improve the issue of unclear boundaries.
  10. Conclusion: It would be beneficial to mention potential future steps or further research directions stemming from this study.
Comments on the Quality of English Language

No problem to read this manuscript.

Reviewer 3 Report

Comments and Suggestions for Authors

Dear Editor and Authors,

Thank you for asking me to review this manuscript titled “Influencing Factors on Intersegmental Identification Adequacy in Segmentectomy with Intraoperative Indocyanine Green (ICG) Intravenous Administration” by Dr. Ueno and his colleagues from the Nagoya University Graduate School of Medicine in Nagoya, Japan.

In this single institution, retrospective analysis the authors present their experience with utilizing indocyanine green (ICG) to identify the intersegmental plain during robotic or VATS segmentectomy. The authors ligate the vessels to the specific segment and then administer ICG intravenously which due to no perfusion of the segment allows them to identify the boundaries of the segment when IR is used.

A total of 111 patients undergoing either RATS or VATS anatomical segmentectomy were collected and analyzed over a three year period. They concluded that patient specific factors such as DLCO were a predictive factor of unclear segmental boundaries.

This is an interesting manuscript describing and providing information regarding a novel technique. It is well written in overall good quality language and requires only minor English editing. It is well illustrated and presented with informative figures and clear tables.

I do have some comments that need addressing:

1.       How were the data for the patients collected (this is not mentioned in the manuscript); were they mined from a departmental research database, from a patient chart review or via another data source? How complete were the data? Were there any missing variables?

2.       Did the authors perform a sample size calculation to establish if the comparisons made between the clear and unclear boundary groups were significant? The number of patients in the unclear boundary group is small (7) so this is a significant limitation in terms of providing reliable comparisons!!

3.       Why was a multivariable analysis done to establish associated factors related to having an unclear boundary?

4.       Was surgical experience a factor in determining the surgical / segmental plains? Was an analysis by surgeon and experience performed?

5.       In terms of the lung cancer histology variable, why was not specific types given (i.e. adenoCa, Squamous, ect!)

6.       Where all lesions peripherally located?

7.       The statistical analysis section needs more explanation. How was ROC calculated?

In conclusion, this is an interesting study and I am positively inclined towards it. However, it has some issues that need addressing prior to acceptance. I await the revision and wish the authors good luck.

Kind regards,  

Comments on the Quality of English Language

The manuscript needs some professional language editing or a good revision/review by a native English speaker.

Round 2

Reviewer 2 Report

Comments and Suggestions for Authors

I am satisfied with the modifications from the Authors as per my comments in this revision. The quality and presentation of this revised manuscript has been improved a lot.

Comments on the Quality of English Language

I have no problem to understand this manuscript.

Reviewer 3 Report

Comments and Suggestions for Authors

Dear Editor and Authors,

Thank you for asking me to re-review this revised manuscript. The authors have addressed the commentary made adequately and I am now happy to recommend its publication.

Kind regards,

Comments on the Quality of English Language

It has been edited by a professional service.